# Gender Disparities in Epidemiology, Treatment, and Outcome for Head and Neck Cancer in Germany: A Population-Based Long-Term Analysis from 1996 to 2016 of the Thuringian Cancer Registry

**DOI:** 10.3390/cancers12113418

**Published:** 2020-11-18

**Authors:** Andreas Dittberner, Benedikt Friedl, Andrea Wittig, Jens Buentzel, Holger Kaftan, Daniel Boeger, Andreas H. Mueller, Stefan Schultze-Mosgau, Peter Schlattmann, Thomas Ernst, Orlando Guntinas-Lichius

**Affiliations:** 1Department of Otorhinolaryngology, Jena University Hospital, 07747 Jena, Germany; andreas.dittberner@med.uni-jena.de (A.D.); benedikt-friedl@gmx.de (B.F.); 2Department of Radiotherapy and Radiation Oncology, Jena University Hospital, 07747 Jena, Germany; andrea.wittig@med.uni-jena.de; 3Department of Otorhinolaryngology, Südharz Klinikum Nordhausen, 99734 Nordhausen, Germany; jens.buentzel@shk-ndh.de; 4Department of Otorhinolaryngology, Helios-Klinikum Erfurt, 99089 Erfurt, Germany; holger.kaftan@helios-gesundheit.de; 5Department of Otorhinolaryngology, SRH Zentralklinikum Suhl, 98527 Suhl, Germany; Daniel.Boeger@srh.de; 6Department of Otorhinolaryngology, SRH Wald-Klinikum Gera, 07548 Gera, Germany; Andreas.Mueller@srh.de; 7Department of Oromaxillofacial Surgery and Plastic Surgery, Jena University Hospital, 07747 Jena, Germany; Stefan.Schultze-Mosgau@med.uni-jena.de; 8Department of Medical Statistics, Computer Sciences and Data Sciences, Jena University Hospital, 07747 Jena, Germany; Peter.Schlattmann@med.uni-jena.de; 9University Tumor Center, Jena University Hospital, 07747 Jena, Germany; thomas.ernst@med.uni-jena.de

**Keywords:** head and neck neoplasm, disparity, cancer registry, risk, epidemiology, survival, cancer incidence and trends

## Abstract

**Simple Summary:**

Head and neck cancer (HNC) comprises a heterogeneous group of cancers. Not much population-based data has been published on gender disparities related to the incidences between different age groups, subsites, tumor stages, and its effect on therapy decisions. All new HNC cases from Thuringia between 1996 and 2016 were analyzed. The incidence of head and neck cancer still was 4-fold higher in men compared to women. Incidence reached a peak for men between 60–64 years, where the incidence increased with older age in women. Male gender, higher tumor stage and subsite (worst: hypopharyngeal cancer) still had a major negative impact on the survival of the patients. Treatment decisions were different between male and female patients, especially in older patients with a tendency to less aggressive therapy. Putting all patients together, there probably was no improvement in survival beyond changes in treatment over the observation period from 1996 to 2016.

**Abstract:**

This study determined with focus on gender disparity whether incidence based on age, tumor characteristics, patterns of care, and survival have changed in a population-based sample of 8288 German patients with head neck cancer (HNC) registered between 1996 and 2016 in Thuringia, a federal state in Germany. The average incidence was 26.13 ± 2.89 for men and 6.23 ± 1.11 per 100,000 population per year for women. The incidence peak for men was reached with 60–64 years (63.61 ± 9.37). Highest incidence in females was reached at ≥85 years (13.93 ± 5.87). Multimodal concepts increased over time (RR = 1.33, CI = 1.26 to 1.40). Median follow-up time was 29.10 months. Overall survival (OS) rate at 5 years was 48.5%. The multivariable analysis showed that male gender (Hazard ratio [HR] = 1.44; CI = 1.32 to 1.58), tumor subsite (worst hypopharyngeal cancer: HR = 1.32; CI = 1.19 to 1.47), and tumor stage (stage IV: HR = 3.40; CI = 3.01 to 3.85) but not the year of diagnosis (HR = 1.00; CI = 0.99 to 1.01) were independent risk factors for worse OS. Gender has an influence on incidence per age group and tumor subsite, and on treatment decision, especially in advanced stage and elderly HNC patients.

## 1. Introduction

Head and neck cancer (HNC) is a heterogeneous group of tumor entities covering several anatomical subsites in the head and neck region. These entities differ greatly in terms of etiology, risk factors, histology, and therapeutic management [1]. Most HNC are still caused by noxious agents like tobacco and alcohol use. However, the increase in incidence of oropharyngeal cancer is caused by persistent infection with human papillomavirus (HPV) [2,3]. In some countries, where tobacco use has declined, the incidence of tobacco-related non-oropharyngeal HNC decreases. Population-based data from larger registries suggest that the overall prognosis of HNC has slightly improved since the 1990s. Overall survival especially for HPV-negative HNC still is poor. Besides HPV status, gender, tumor subsite and stage influence survival [4]. About half of the patients are still diagnosed at advanced stage [5,6]. Since 2000s, curative therapy concepts for early-stage disease are characterized by monotherapeutic concepts (surgery alone or radiotherapy alone), whereas locally-advanced stages are treated by multimodal approaches [1]. Chemotherapy regimens as part of concurrent radiochemotherapy are dominated by the use of a platin derivate. Taxane played a role in induction chemotherapy. Cetuximab was the only licensed monoclonal antibody, established in systematic treatment concepts for recurrent or metastatic disease, but played a minor role as substitute for chemotherapy in curative therapy concepts [7].

Actual epidemiologic population-based studies on head and neck cancer are still sparse, nation-wide head and neck cancer registry data is only available from some countries (for instance, [8,9]). Important data comes from programs covering parts of the population (United States: Surveillance, Epidemiology, and End Results [SEER] Program and National Cancer Database [NCDB], case coverage 30–70%; Europe: EUROCARE-5 study; case coverage unclear) [5,10,11], but analyses of gender disparities are still a neglected topic.

Some years ago, we published data on epidemiology and prognosis of HNC cancer diagnosed in 1996 to 2011 covering the federal state Thuringia in Germany [12]. We here present updated data on survival and prognostic factors for head and neck cancer from 1996 to 2016 in Germany with special focus on gender-related incidence rates and treatment trends in clinical routine beyond clinical trials.

## 2. Results

### 2.1. Patient’s Characteristics, Tumor Characteristics and Therapy Strategies

We identified 8288 cases fulfilling the selection criteria. The number of cases per year varied from 313 to 482. About four out of five patients were male (6540 male; 78.9%). The median age at diagnosis was 60 years (range: 12–100). Female patients (64.1 ± 14.1 years) were older than male patients (60.1 ± 11.1 years; *p* < 0.001). Whereas only about one third of the male patients were ≥65 years, half of the female patients were ≥65 years at time of diagnosis (Figure 1). Appendix A shows the distribution of the cases according to year of diagnosis, registry region and other patients’ characteristics. As shown in Table 1, tumors of the oral cavity, oropharynx and larynx represented about two thirds of all head and neck tumors. Four out of five patients had a squamous cell carcinoma. About two thirds of the male patients and more than half of the female patients were diagnosed at advanced stage (stage III/IV; Figure 1). Higher T classification was associated with higher N classification (Appendix A). The relation between tumor stage and tumor subsite is summarized in Appendix A. The proportion of higher N classification is increasing with higher T classification. Lip cancer and laryngeal cancer were the only two subsites with a relatively high amount of patients diagnosed in stage I. Hypopharyngeal cancer was almost exclusively diagnosed in stage IV. Three-hundred-and-sixty-two (362) patients (4.4%) had distant metastasis (M+) at primary diagnosis (Appendix A). The most frequent sites for distant metastasis were the lung, lymph nodes outside the head and neck region, liver and bone. Surgery as single modality treatment was the most frequent treatment option, followed by the combination of surgery and adjuvant radiotherapy or radiochemotherapy (cf. Table 1). Most patients in stage I/II received a single modality treatment. In stage III/IV, the vast majority of patients received a multimodal therapy (Appendix A). With higher age, the proportion of monotherapy instead of multimodal therapy in stage III/IV increased (Appendix A). The majority of stage III/IV patients with age of ≥80 years received a monotherapy. The relative proportion of monotherapy in the oldest age group was higher in older female than in male patients.

### 2.2. Epidemiology: Crude Incidences of Patients’ and Tumor Characteristics between 1996 and 2016

Seven-thousand-five-hundred-and-two (7502) patients (90.5%) lived in Thuringia and were included into the incidence calculation. The average incidence for all patients over the entire observation period was 15.98 ± 1.81 per 100,000 population per year (men: 26.13 ± 2.89; women: 6.23 ± 1.11; Appendix A). A significant but small increase of the incidence between 1996 and 2016 was exclusively observed for females in the age group of 55–59 years (Relative risk [RR] = 1.20; 95% confidence interval [CI] = 1.06 to 1.35). The crude incidence for Thuringia corresponded to an ESR incidence rate of 12.68 ± 1.14 (Appendix A). The incidence peak for male patients was reached with 60–64 years of age (63.61 ± 9.37) and was decreasing with older age, independent of the year of diagnosis (Figure 2). In contrast, the incidence increased over age without a peak in female patients reaching the highest incidence with ≥85 years of age. Oropharyngeal cancer had the highest incidence from all subsites independent of the gender (4.23 ± 0.93; Figure 3; Appendix A). In male patients, the incidence of cancer of the oral cavity (RR = 1.06; CI = 1.01 to 1.10), oropharyngeal cancer (RR = 1.17; CI = 1.12 to 1.21), and nasal/paranasal sinus cancer (RR = 1.24; CI = 1.08 to 1.42) slightly increased in the observation period. Female patients showed a weak increase of the incidence over time for cancer of the oral cavity (RR = 1.14; CI = 1.06 to 1.23) and oropharyngeal cancer (RR = 1.11; CI = 1.02 to 1.21; Appendix A). The incidence of lip cancer decreased over time in male patients (RR = 0.78; CI = 0.69 to 0.90). For both genders, a significant decrease of incidence in any other subsite was not seen. Incidence was highest in stage IV (6.67 ± 1.15), with a more prominent effect in men (11.44 ± 1.89) compared to women (2.06 ± 0.66; Figure 3; Appendix A). Unstaged cases decreased over the observation period (RR = 0.91; CI = 0.86 to 0.95). Instead, mainly the incidence cases classified as stage IV increased slightly in male patients (RR = 1.10; CI = 1.06 to 1.13) and stronger in female patients (RR = 1.21; CI = 1.12 to 1.30).

### 2.3. Treatment Strategies

The predominant treatment strategies were in descending order (cf. Table 1): Surgery alone (26.5% of the cases), surgery with adjuvant radiochemotherapy (21.2%), surgery with adjuvant radiotherapy (21.0%), and definitive radiochemotherapy/radioimmunotherapy (11.9%). During the period studied, several trends concerning the treatment strategies were encountered (Appendix A): The relative frequency of radiotherapy as single therapy decreased (RR = 0.83; CI = 0.78 to 0.89). Radiochemotherapy/radio-immunotherapy increased slightly (RR = 1.08; CI = 1.02 to 1.13). Surgery increased as a single modality (RR = 1.07; CI = 1.03 to 1.10), but also as primary treatment in combination with radiochemotherapy/radio-immunotherapy (RR = 1.05; 1.01 to 1.09). Overall, monotherapeutic concepts showed no change over time (RR = 1.00; CI = 0.96 to 1.04), whereas multimodal concepts increased (RR = 1.33, CI = 1.26 to 1.40). The use of chemotherapy/biologicals as part of the treatment concept increased (RR = 1.16; CI = 1.12 to 1.19), which is most evident for the use of cetuximab (RR = 2.06; CI = 1.85 to 2.29).

### 2.4. Tumor Recurrence and Overall Survival

The median follow-up time of all patients was 29.10 months (mean: 46.08; 95% CI: 44.07–46.08 months). The median follow-up time of all patients alive at the end of the observation period was 50.43 months (mean: 59.64; 95% CI: 58.04–61.25 months). No recurrence was observed in 6958 patients (84%). One-thousand-three-hundred-and-thirty patients (16%) developed a recurrence. More data on the recurrence localization is shown in Appendix A. Four-thousand-four-hundred-and-eighty-two patients (54.1%) died during follow-up. 3806 patients (45.9%) were alive. Median overall survival time was 54.3 months (CI: 51.8–58.7 months). Overall survival rate at 2 and 5 years for all patients was 65.4% and 48.5%, respectively. Male gender, higher TNM staging, but not the date of primary diagnosis were negative prognostic factors for overall survival (Figure 4, Appendix A). Appendix A shows the 5-year overall survival rates related to patient’s age and tumor stage separately for all patients, male and female patients. Over time, there was no clear trend for better overall survival in any age group. Overall survival seemed to rather decrease in patients ≥85 years of age. Both male and female stage IV patients showed better 5-year overall survival over the observation time. The multivariable analysis was based on a Cox model including baseline data, tumor characteristics, and treatment concepts. The multivariable analysis revealed that male gender (Hazard ratio [HR] = 1.44; CI = 1.32 to 1.58), tumor subsite (worst hypopharyngeal cancer: HR = 1.32; CI = 1.19 to 1.46), and tumor stage (stage IV: HR = 3.40; CI = 3.00 to 3.85) were independent significant negative risk factors for worse overall survival (all *p* < 0.05; Table 2). Furthermore, the treatment concept had significant influence on the outcome. Putting all patients together, there probably was no improvement in survival beyond changes in treatment over the observation period from 1996 to 2016 (HR = 1.00; CI = 0.99 to 1.01).

## 3. Discussion

Global cancer burden including HNC is increasing, with population aging and population growth as major contribution factors [13]. Thuringia is a federal state in Germany with actually 2.1 million habitants. As >97% of the HNC patients in Thuringia were Caucasian, ethnic disparities as an influence factor did not play a role [14,15]. With an absolute average incidence for all patients over the analyzed period of 15.98 per 100,000 population per year, head and neck cancer is an important health care burden. This incidence is in the lower range of other industrial countries varying from about 15 to 43 per 100,000 [8,9,16]. The overall incidence of HNC did not change over time from 1996 to 2016 in Thuringia. In contrast, the overall incidence in some other industrial countries like the United States is decreasing [17]. Despite declining smoking prevalence, the incidence of nasopharynx, hypopharynx and larynx cancer remained quite stable in many European countries, while those of oropharynx and oral cavity increased [5]. Some countries in Europe observe at least a decline in laryngeal cancer in male patients [8]. This is attributed to decreasing smoking habits, but it has to be taken into account that it takes generally more than 20 years between the first genetic event and the eventual clinical detection of HNC [18]. The clinical cancer registries in Germany, hence also in Thuringia, do not include data on smoking habits. Thuringia is a state with a high smoking prevalence in head and neck cancer patients in Germany [19]. The smoking prevalence was 38% (43% in male patients, 62% in male patients with laryngeal cancer) for the years 2009 to 2011 in Thuringia [20]. This might explain why we could not find a decline in male laryngeal cancer patients in Thuringia.

The increase of oropharyngeal and oral cavity cancer in many countries is mainly explained by the increase of human papillomavirus (HPV)-related cancer [21]. Unfortunately, the HPV status was not recorded in the Thuringian cancer registries until 2019 because afore the HPV status did not influence the TNM classification. The rate of HPV+ oropharyngeal cancer in Europe (about 31.1%) is much lower than in United States (about 59.3%) [21,22]. Within the East German states, still about 60% of the patients with oropharyngeal cancer are smokers [19]. Hence, smoking still is in some parts of Germany, including Thuringia, the main driver for oropharyngeal cancer. This might be the reason why we observed a lower increase of oropharyngeal cancer in Thuringia than in other countries. As improved survival is expected for patients with HPV+ oropharyngeal cancer, this factor might also explain why we did not observe an increase of survival in patients with oropharyngeal cancer.

Several important gender differences were observed. Head and neck cancer is still dominated by male patients (range: 74–82% of all patients, no significant trend over time). The proportion of stage IV cases increased for both genders and might be an effect of better diagnostics and better staging, but does not clearly explain why the increase of stage IV cases over time was higher for females. Female patients were older than male patients. The incidence peak for male patients was reached with 60–64 years of age while incidence rates were decreasing with older age. In contrast, incidence was continuously increasing with advancing age in women, reaching the highest incidence with ≥85 years of age. In addition, there was no increase of incidence for any age group in both genders, except for a small group in females 50–59 years of age. Partly, the different proportions of female patients in the different subsites can explain incidence disparities. The proportion of women compared to men was high for HNC subsites more typical for older age (lip cancer: 33%; nasal/paranasal cancer: 37%; salivary gland cancer: 41.2%), but low for subsites associated to relatively lower age (oropharyngeal cancer: 19%; hypopharyngeal cancer: 11%; laryngeal cancer: 9%). Furthermore, treatment decisions seem to be different between men and women, especially in higher stage and older age. The relative proportion of monotherapy, especially in advanced stage, in the oldest age group was higher in older female than in male patients. A previous study has also shown that multimodality treatment is less frequently applied with increasing age [23]. Age as a single parameter is no contraindication for multimodal therapy. The present registry-based study does not allow analyzing reasons against non-standard treatment in advanced HNC. Comorbidity, which increases with age, can be a contraindication for multimodal therapy [20]. Unfortunately, German cancer registries do not contain comorbidity data. Furthermore, it is assumed that patients’ wishes in older age, their social context and support, and possibly prejudices or personal preferences of health care givers are different to younger patients [23]. Such soft factors might be different between men and women. A previous U.S. American study analyzing data of the SEER registry from 2000 to 2015 found an undertreatment of women with locally advanced HNC [24]. Furthermore, another SEER analysis including data from 1985 to 2015 showed that women are less likely than men to receive definitive radiochemotherapy as opposed to definitive radiotherapy [25]. It can be hypothesized that other unmeasured factors, including an implicit physician bias and variation in sex-different patient treatment goals, may contribute to the lower utilization of intensive therapy in female patients [24].

Nevertheless, overall survival still is significantly worse in men compared to women with HNC. For many cancers entities, age of female tumor patients at primary diagnosis is lower than the age of men, which might contribute to the better overall prognosis [26]. The lower use of tobacco and thus lower incidence of comorbidity likely contributes to the survival advantage of female tumor patients over men [27]. Other reasons for the survival advantage are not known. Genomic analyses could not yet find significant sex disparities in HNC driver genes [28].

In conclusion, the presented large registry-based study did not find a significant improvement of survival for HNC patients in Thuringia despite an increase of more multimodal therapy over time. Several gender disparities in patients’ characteristics, tumor characteristics and treatment were found. These disparities can only be partly explained. Reasons for the above disparities need further prospective investigation and probably adaptation of guidelines to correct such inequalities. Furthermore, it would be helpful for better understanding of treatment decisions, if the cancer registries would also include better data on comorbidity and treatment decision making.

## 4. Material and Methods

The Ethics Committee of the Jena University Hospital approved the study (IRB No. 3204-07/11). The Ethics Committee waived the requirement for informed consent of the patients because the study had a non-interventional retrospective design and all data were analyzed anonymously. All procedures of the study involving human participants were in accordance with the Declaration of Helsinki (1964) and its later amendments or comparable ethical standards. All authors had access to the study data and reviewed and approved this study.

### 4.1. Patients

Data of the Thuringian cancer registry database was used. The population-based Thuringian cancer registry covers all cancer cases of the federal state Thuringia. It collects data from the five Thuringian cancer registers (Nordhausen, Gera, Suhl, Jena and Erfurt). Thuringia has a population of about 2 million people. The registry has data of about 98% of all head and neck cancer patients in Thuringia [12]. All new patients with head and neck cancer registered between 1996 and 2016 were included. The tumors were classified according to the International Classification of Disease for Oncology, third edition, first revision (ICD-O-3) [29]. Based on the ICD-O-3 codes, cancers were divided into the following subsites: lip, oral cavity, nasopharynx, oropharynx, hypopharynx, larynx, salivary glands, and nose/paranasal sinus. Cases of head and neck cancer that could not be distinguished by a specific site because of invading more than one subsite were grouped as “unspecified”. Patients who were treated for recurrent disease only, skin cancer, metastases of other entities in the head and neck region, and carcinoma of unknown primary were excluded. Duplicate records of patients were removed.

Extent of the disease was classified by pathological stages (pTNM) for all cases with a surgical resection. Otherwise, clinical stages (cTNM) were used. Staging was defined by the AJCC Cancer Staging Classification, 7th edition (2010). T or N classification were not clearly defined in all cases. Therefore, stage grouping was not possible for all cases. Treatment was defined to be the first course of cancer-specific therapy of the primary tumor. Subsequent treatment for recurrent disease was not included in this definition of treatment.

### 4.2. Statistical Analysis

Statistical analyses were performed using IBM SPSS version 25.0 statistical software for Windows (Chicago, IL, USA) and SAS 9.4 (SAS Institute Inc., Cary, NC, USA). Incidences were calculated per 100,000 inhabitants based on yearly estimates of Thuringian residents (Statistical Bureau of Thuringia: [30]). All rates were age-standardized to the European standard population 2013 (European Standardized Rates–ESR) [31]. Overall survival (OS) was calculated by the Kaplan–Meier method. Differences of survival were compared by the log-rank test. Multivariable analysis was performed using the Cox proportional hazards model to estimate the hazard ratio (HR) with a confidence interval (CI) of 95% for overall survival. Poisson regression models with log link were performed to conduct an analysis over time. Here, the dependent variable was the number of cases and the logarithm of the population at risk was taken as an offset. Time from 1996 was taken as an independent variable. Relative risks (RR) with a CI of 95% are reported for a time period of 5 years. Concerning the age groups, the RR were related to the group of patients ≤60 year of age as reference group. Time trend analysis was conducted for age strata consisting of 5-year groups, e.g., 55–59 over the entire period from 1996 to 2016. For all statistical tests, significance was two-sided and set to *p* < 0.05.

## 5. Conclusions

We have demonstrated that, overall, HNC survival has not improved in the observed time despite relevant therapy changes. There are many gender disparities: Male patients still have worse survival. Incidence peaks and distribution to HNC subsites are different between men and women. Therapy decisions seem to depend on the age of the patients but also on gender. These differences in decision making between men and women have to be better understood.

## Figures and Tables

**Figure 1 cancers-12-03418-f001:**
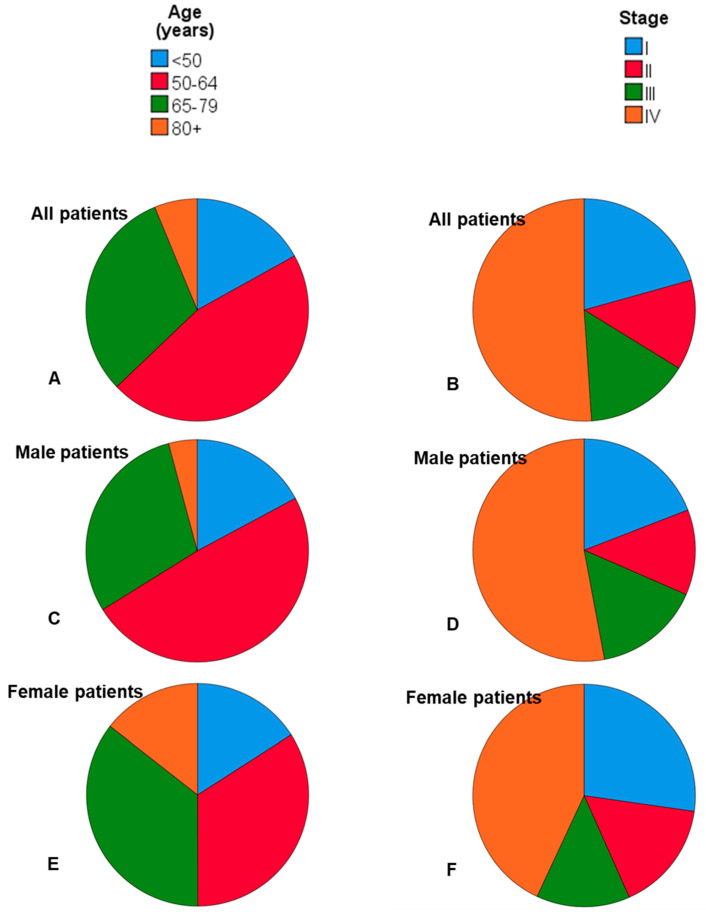
Distribution of age (**A**,**C**,**E**) and tumor stage (**B**,**D**,**F**) in all patients (**A**,**B**), male patients (**C**,**D**), and female patients (**E**,**F**).

**Figure 2 cancers-12-03418-f002:**
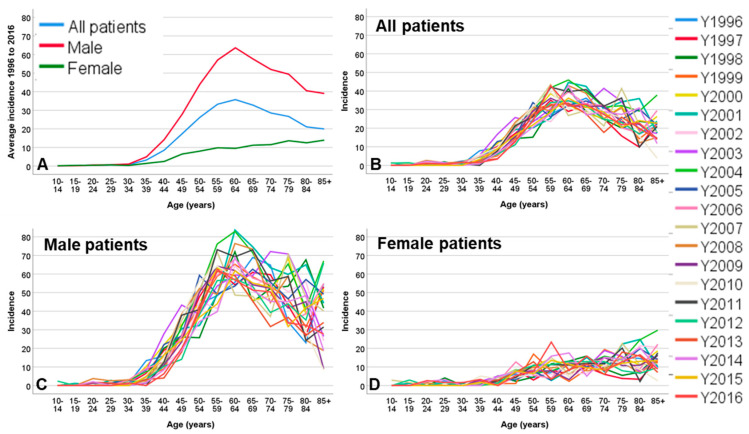
Incidence rates of head and neck cancer per 100,000 population in Thuringia per age group. (**A**) Average incidence rates from 1996 to 2016 for all patients, male and female patients. (**B**–**D**) Incidence rates for each year and age group for all patients (**B**), male patients (**C**), and female patients (**D**).

**Figure 3 cancers-12-03418-f003:**
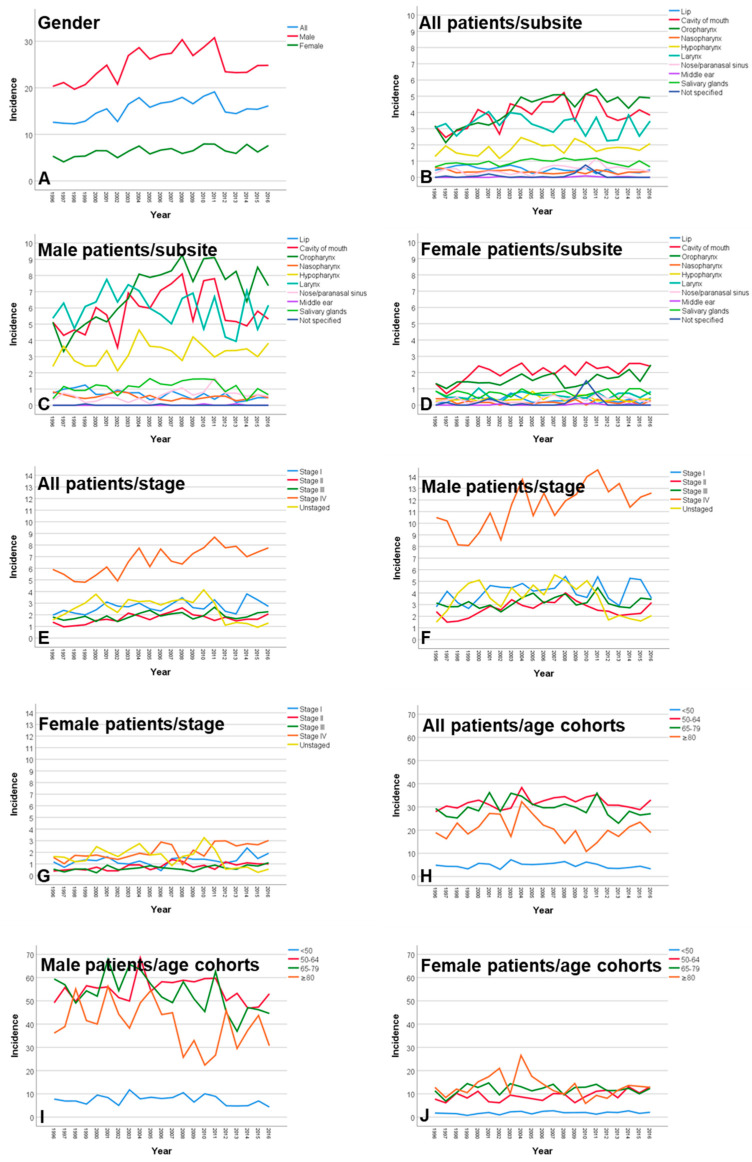
Incidence rates of head and neck cancer per 100,000 population in Thuringia per year from 1996 to 2016. (**A**) Related to gender. (**B**) Related to subsite in all patients. (**C**) Related to subsite in male patients. (**D**) Related to subsite in female patients. (**E**) Related to tumor stage in all patients. (**F**) Related to tumor stage in male patients. (**G**) Related to tumor stage in female patients. (**H**) Related to age groups in all patients. (**I**) Related to age groups in male patients. (**J**) Related to age groups in female patients.

**Figure 4 cancers-12-03418-f004:**
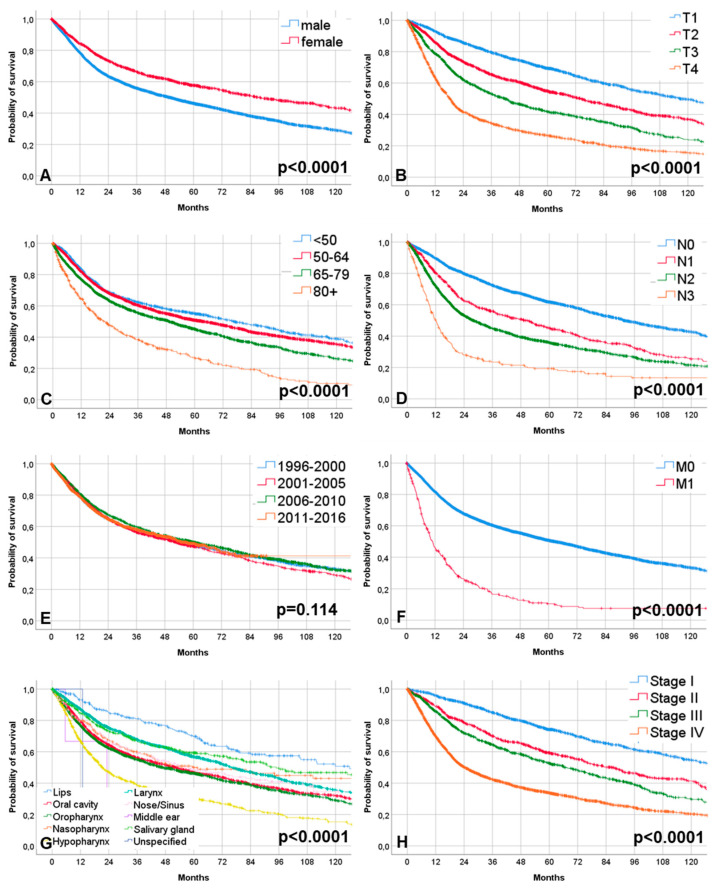
Kaplan–Meier curves of the overall survival. (**A**) Related to gender. (**B**) Related to T classification. (**C**) Related to age groups. (**D**) N classification. (**E**) Related to periods of the year of diagnosis. (**F**) Related to M classification. (**G**) Related to tumor subsite. (**H**) Related to tumor stage.

**Table 1 cancers-12-03418-t001:** Tumor and therapy characteristics.

Tumor Site	Frequency (N)	%
All	8288	100
Lips	275	3.3
Cavity of mouth	2116	25.5
Oropharynx	2240	27.0
Nasopharynx	191	2.3
Hypopharynx	941	11.4
Larynx	1737	21.0
Nose and paranasal sinus	246	3.0
Middle ear	7	0.1
Salivary glands	490	5.9
Not classifiable *	45	0.5
T classification		
T1	2019	24.4
T2	1836	22.2
T3	1303	15.7
T4	1941	23.4
Tx	1189	14.3
N classification		
N0	3352	40.4
N1	783	9.4
N2	2457	29.6
N3	285	3.4
Nx	1411	17.0
M classification		
M0	7212	87.0
M1	362	4.4
Mx	714	8.6
Stage (AJCC 7th edition 2010)		
I	1435	17.3
II	911	11.0
III	1052	12.7
IV	3546	42.8
Unstaged	1344	16.2
Stage (SEER)		
Localized	3294	39.7
Regionalized	3276	39.5
Distant	360	4.3
Unstaged	1358	16.4
Therapy		
No therapy **	434	5.2
Radiotherapy alone	567	6.8
Radiochemotherapy or radioimmunotherapy	985	11.9
Surgery alone	2193	26.5
Surgery and chemotherapy	92	1.1
Surgery and radiotherapy	1738	21.0
Surgery and radiochemotherapy	1755	21.2
Chemotherapy or immunotherapy alone	83	1.0
Radiochemotherapy and immunotherapy	35	0.4
Surgery, radiotherapy, chemo- and immunotherapy	78	0.9
Unknown	328	4.0
Histology		
Squamous cell carcinoma	7034	84.9
Adenocarcinoma	200	2.4
Other carcinoma	811	9.8
Other neoplasia	243	2.9

* not classifiable to a subsite, overlapping several subsites; ** best supportive care not evaluated; AJCC = American Joint Committee on Cancer; SEER = Surveillance, Epidemiology, and End Results Program.

**Table 2 cancers-12-03418-t002:** Multivariable Cox regression of risk factors for head and neck cancer overall survival including the year of diagnosis.*

Factor	Parameter	HR	Lower 95% CI	Upper 95% CI	*p*
Age	≤60 years	1	Reference		
	>60 years	1.405	1.316	1.502	**<0.0001**
Gender	Female	1	Reference		
	Male	1.444	1.317	1.582	**<0.0001**
Tumor site	Oropharynx	1	Reference		
	Lip	0.969	0.757	1.241	0.8043
	Oral cavity	1.298	1.188	1.419	**<0.0001**
	Nasopharynx	0.657	0.519	0.832	**0.0005**
	Hypopharynx	1.321	1.192	1.465	**<0.0001**
	Larynx	1.045	0.944	1.156	0.3983
	Nose/paranasal	1.022	0.822	1.271	0.8458
	Salivary gland	0.915	0.763	1.096	0.3326
	Not classifiable	8.018	1.124	57.188	**0.0378**
Stage	I	1	Reference		
	II	1.801	1.572	2.064	**<0.0001**
	III	2.258	1.967	2.593	**<0.0001**
	IV	3.402	3.005	3.851	**<0.0001**
Therapy	Surgery alone	1	Reference		
	Surgery and radiotherapy	0.752	0.674	0.840	**<0.0001**
	Surgery and chemotherapy	1.329	0.989	1.787	0.0596
	Surgery and radiochemotherapy	0.826	0.739	0.924	0.0008
	Surgery, radiotherapy, chemo- and immunotherapy	0.858	0.611	1.205	0.3771
	Radiotherapy alone	1.932	1.675	2.229	**<0.0001**
	Radiochemotherapy and immunotherapy	1.623	1.062	2.478	0.0251
	Radiochemotherapy or radioimmunotherapy	1.382	1.219	1.566	**<0.0001**
	Chemotherapy or immunotherapy alone	3.145	2.345	4.220	**<0.0001**
	No therapy	2.792	2.370	3.290	**<0.0001**
Year of diagnosis		1.000	0.994	1.006	0.9460

HR = Hazard ratio; CI = confidence interval; * significant *p*-values in bold.

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
