# Peer review of "Gender Disparities in Epidemiology, Treatment, and Outcome for Head and Neck Cancer in Germany: A Population-Based Long-Term Analysis from 1996 to 2016 of the Thuringian Cancer Registry"

_cancers, 2020, doi:10.3390/cancers12113418_

Round 1

Reviewer 1 Report

Dittberner et al. discuss gender disparities in the context of head and neck cancer treatment and outcomes in a German state. Overall these type of population studies can be useful and US data are fairly limited due to inherent reporting problems with SEER and NCDB so there is value in the current study overall.

Positive comments

  1. Well written overall. Thorough data presentation.
  2. Incidence data are useful to the readership and well presented.
  3. Statistical analysis is robust.

Negative comments

  1. Time span of 20 years makes it difficult to adjust outcomes to temporal shifts related to smoking and other demographic shifts. Not really fixable but something that can impact results. Partially addressed in Discussion with respect to high smoking prevalence.
  2. As would be expected there are multiple interacting factors. Gender interacts with age, age interacts with treatment intensity, treatment intensity when combined with stage drives survival. It is therefore very difficult to tease out all of these interactions with the number of patients in the study.
  3. HPV associated oropharyngeal cancer is not addressed. I expect that registry data lack p16 or HPV status, certainly for dates prior to 2010. The authors should make an effort to correlate shifts in disease incidence and tobacco exposure across sites as a surrogate for the epidemiological shift we know is happening in OPSCC. Is this possible with the data available?
  4. Mixing of sites and histologies makes data interpretation somewhat challenging. It would be useful if the authors conducted a subset analysis using the most common site (i.e. oral cavity, oropharynx, larynx) of SCC origin in parallel and evaluated the gender question in this more homogeneous cohort. Otherwise, I am afraid it is a little too hard to interpret the gender differences in a way that is useful and makes sense.

Questions:

  1. What are “lymph nodes beyond the neck”? Does this refer to mediastinal nodes, axillary nodes, supraclavicular nodes? How is this determination made especially for low level IV and level V vs supraclavicular nodes? Alternatively paratracheal nodes in the central compartment for laryngeal cancer vs upper mediastinal nodes?
  2. I’m confused. 84% of patients had no recurrence. However, 54% of patients died during follow up with a median follow up of 50 months. That means that over 30% of patients without a recurrence died within 5 years of cancer treatment……this seems very high, especially since we are talking about individuals in their early 70s. These numbers require clarification and some effort must be made to adjust this for life expectancy in the general population during this time period for this region.

Author Response

Additional response to the comments of the editors:

  1. Editorial team

We thank the editorial team for their additional questions.

3.1. Please provide a graphical abstract.

Answer 3.1.: Done.

3.2. In the references list, please add the first ten authors then use et al.

Answer 3.2.: Done.

3.3. There are much overlaps in "Section 4. Material and Methods", the overlaps are highlighted in the attached manuscript version, please rewrite these overlaps.

Answer 3.2.: We rewrote parts of the Material and Methods section. The section does not contain relevant original contents. The section describes standard methods for population-based cancer registry analyses.

Orlando Guntinas-Lichius

For all authors

Jena, 13-Nov-20

Reviewer 2 Report

This well-written manuscript by Dittberner and Coworkers is dealing with population-based data from the Thuringian cancer registry. The authors have analyzed more than 8000 head and neck cancer (HNC) for the years 1996-2016 regarding gender disparities in epidemiology, treatment, and outcome.

The study contains an impressive number of cases. Overall, the work was well planned and executed.

I have two suggestions:
 1.  Even if the individual diseases are not the focus of the study, an additional supplementary table about their distribution in percent would be interesting.
 2.  Figures 2 and 3 are hard to identify. Perhaps it is possible to increase the figure size.
